# Size Measurement and Segmentectomy Resection Margin of Early-Stage Lung Adenocarcinoma Manifesting on Virtual 3D Imagery and Pathology: A Pilot Correlation Study

**DOI:** 10.3390/jcm11206155

**Published:** 2022-10-19

**Authors:** Ching-Min Lin, Hui-Chun Tai, Ya-Fu Cheng, Pei-Cing Ke, Chia-Chi Liu, Bing-Yen Wang

**Affiliations:** 1Division of Thoracic Surgery, Department of Surgery, Changhua Christian Hospital, Changhua 500209, Taiwan; 2Department of Pathology, Changhua Christian Hospital, Changhua 500209, Taiwan; 3Department of Post-Baccalaureate Medicine, College of Medicine, National Chung Hsing University, Taichung 402202, Taiwan; 4Institute of Genomics and Bioinformatics, National Chung Hsing University, Taichung 402202, Taiwan

**Keywords:** adenocarcinoma, tumor size, resection margin, segmentectomy, pre-operative simulation, 3D reconstruction

## Abstract

Background: The objective of our study was to assess if 3D reconstructed images could be extrapolated to reflect pathologies, as evaluated by early-stage lung adenocarcinoma tumor size and simulated segmentectomy resection margin. Methods: Retrospectively selected patients (*n* = 18) who underwent segmentectomy at Changhua Christian Hospital between 2012 and 2018 and then had pulmonary 3D reconstruction using Ziostation2 were included in our study. Tumor size and simulated segmentectomy resection distance on a 3D model were measure and compared to pathology. Results: Both tumor size and segmentectomy resection margin showed positive correlations between 3D image measurements and pathological measurements. The resection margin showed a stronger correlation and was beneficial in pre-operative planning. Conclusions: A 3D reconstructed model aided understanding of pulmonary anatomy, prompting confidence in surgical approaches and ensured segmentectomy outcome success. Regardless of age and pulmonary function, 3D simulation can accurately mimic segmentectomy, making it a simple, effective and feasible pre-operative planning tool.

## 1. Introduction

Since the publication of a randomized controlled trial performed by the Lung Cancer Study Group in 1995, lobectomy with adequate lymph node dissection was considered the gold standard treatment for early-stage non-small cell lung cancer (NSCLC) [1]. This concept has been challenged and is currently opposed. Nowadays, the widespread use of thin-slice computed tomography (CT) imaging in clinical practice and the application of lung cancer screening programs have resulted in high incidence of lung nodules in patients of younger age, with tumors of smaller size and with cancer diagnoses at earlier stages of presentation [2]. Under equivalent oncologic outcomes, sublobar resection, especially segmentectomy, is now favored because it preserves more lung function, has less perioperative morbidities and increases the possibility of surgery for secondary primary lung cancer [3,4,5,6,7].

Accurate size measurement of pulmonary nodules is a prerequisite for accurate nodule management because staging and treatment options are based on size. However, tumor size discrepancies exist between radiological measurement and pathological measurement [8,9,10,11]. This disparity is due to different pulmonary status: chest CT is conducted under lung expansion whereas surgical resection that yields pathological specimens is performed in a collapsed lung state. Resected lung tissue becomes flattened due to deflation and blood drainage [9]. Moreover, pathology specimen processing, including formalin fixation and slide preparation processes, may change pathological tumor size [10]. This tumor shrinkage can cause downstaging in 3–10% of specimens in NSCLC11. 

Adequate clear resection margins are key to curative intent in early NSCLC. The latest National Comprehensive Cancer Network guideline for NSCLC recommends a resection margin distance greater than or equal to the tumor size when sublobar resection is performed [12]. With segmentectomy becoming a mainstay, complete pre-operative understanding of pulmonary anatomy is vital. However, anatomical variations are difficult to map out on CT images. The development of multi-detector CT has enabled three-dimensional (3D) image reconstruction of lung structures, including segmental divisions, vascular arborization and bronchial anatomy. The reconstructed image can overcome challenges with conventional CT images and be applied to aid case selection and pre-operative planning. 

The purpose of this pilot study was two-fold. First, we assessed the correlation of tumor size correlation with lung adenocarcinoma pathological measurement and 3D image measurement. Next, we investigated whether 3D image reconstruction software was a feasible operative planning tool, as measured by simulated resection margin. These comparisons can help quantify potential differences between the real and virtual world and provide better management. 

## 2. Materials and Methods

### 2.1. Case Selection 

Retrospective chart review was performed for patients who underwent segmentectomy between 2012 and 2018 at the medical center in Changhua Christian Hospital, Changhua City, Taiwan. The search yielded 200 patients who underwent pulmonary segmentectomy; these patients were matched to our hospital’s radiology database to identify those with pre-resection CT examinations available in the picture archiving and communication system. Our chest CT lung window setting was window width 1400 to 1600 HU and window level −350 to −500 HU. Inclusion criteria were pathologically verified primary lung adenocarcinoma, availability of the most recent chest CT performed within 6 weeks prior to operation, CT diameter <30 mm (T1 stage) as measured on the lung window setting. Exclusion criteria were unavailability of the most recent chest CT images prior to operation, CT nodule size diameter >30 mm as measured on the lung window setting and multiple nodules at the ipsilateral lobe. From the operation notes, we further excluded patient who first underwent wedge resection, then segmentectomy. Finally, if Ziostation2 could not generate 3D reconstruction images or if its simulated segmentectomy yielded incomplete segmentation owing to factors such as the CT slide being too thick (i.e., exceeding 1.25 mm) and bronchi to the resected segment being not fully shown, the case was also excluded. In total, we enrolled 18 patients who were eligible.

### 2.2. 3D Simulation Measurement and Segmentectomy

Operation date was obtained from medical records, then the closest-to-operation date chest CT DICOM images were obtained and imported into Ziostation2 software (Ziosoft, Tokyo, Japan). Using the CT lung resection planning function, 3D images of high precision automatic segmentation of lungs, lung lobes, bronchi and pulmonary vessels were created. Tumor location was manually indicated, then tumor boundary was manually edited to exclude peri-tumor vessels on the 3D models. Maximum tumor diameter was measured on the 3D image. Resection margin was set at 20 mm and simulated segmental resection was based on the bronchi method. We specified ligature marks according to segmentectomy operation records, then auto-division resection was performed. The segmented bronchi were shown as non-colored areas while remaining segments were colored (Figure 1). The surgical margin between the tumor and the closest fissure was automatically shown.

### 2.3. Surgical Technique

Intersegmental plane was performed via the inflation–deflation method. First, the lung on the operating side was deflated for successful dissection and ligation of targeted segment bronchus, artery and vein. Then, the lung was reinflated with 100% oxygen and allowed to fully expand by collateral ventilation via Kokn’s pores. The air in the preserved segments would be carried away by blood or released through intubation whereas the target segment remained inflated [13]. Hence, a clear intersegmental plane was identified. The bronchus was resected using a stapler, then an endo-stapler (EndoGIATM, COVIDIEN Medical, Minneapolis, MN, USA) was introduced and the segmental borders were divided according to the inflation–deflation line.

### 2.4. Pathology Tumor Size and Resection Margin

Upon receiving the resected segmentectomy specimen at our pathology department, tumor size was determined by gross measurement using a standard ruler via naked eye. The specimens were not inflated with air, embedding medium or fixative before gross cutting. The obtained specimen was fixed in 10% neutralized formalin and embedded in paraffin blocks then sections were cut into 3-um thick slices and stained with hematoxylin and eosin stain. Pathological tumor size was confirmed again by microscopic measurement on hematoxylin and eosin slides under a light microscope. If there was a discrepancy in tumor size measurement, microscopic measurement overruled the gross measurement. For this study, the longest pathologic diameter of each adenocarcinoma was extracted from the cancer staging section of the pathology report. Resection margin was measured as the nearest margin regardless of specimen orientation.

### 2.5. Statistical Analysis

Descriptive statistics and categorical variables were tabulated based on medical records. The linear regression equations and coefficient of determination were fitted to a scatter curve of tumor size and resection margin. Pearson’s correlation coefficient was used for the analysis. Univariable and multivariable analyses were performed. All calculations and analyses were performed using SPSS Statistic, version 23.0 (SPSS Inc., Chicago, IL, USA) software. A 2-tail *p*-value < 0.05 was considered statistically significant. Due to our small sample size, *p*-values may not be interpreted as confirmatory but rather descriptive.

## 3. Results

### 3.1. Patient Characteristics

A total of 18 patients fulfilled the inclusion and exclusion criteria and were included in the analysis. Demography of patients is provided in Table 1. There were 7 males (38.9%) and 11 females (61.1%) in this cohort. Their median age was 60 years (IQR 47.50–65.25). Underlying diseases of this cohort include four diabetics (22%), five hypertensives (27.8%), two smokers and one chronic obstructive pulmonary disease patient (5.6%). A pulmonary function test conducted prior to segmentectomy showed median FEV1 of 2.40 (IQR 1.91–2.62), best/predicted percentage of 89.20 (81.25–105.15) and median FEV1/FVC of 78.88. Most of the pulmonary nodules (11/18, 61.1%) were located in the right lung, but there were seven (38.9%) left lung nodules. Subtypes of adenocarcinoma include nine (50%) acinar, four (22.2%) lepidic (formerly known as bronchioloalveolar), three (16.6%) papillary and two (11.1%) minimal invasive adenocarcinomas. Median tumor size, as measured on pathology, was 1.31 cm (IQR 0.87–1.64) whereas virtual 3D measurement was 0.95 cm (IQR 0.60–1.55). Median tumor resection margin on pathology was 1.48 cm (IQR 0.50–3.35) whereas virtual 3D measurement was 1.70 cm (0.98–2.12). This patient cohort’s median length of hospital stay was 5 (IQR 5–7) days.

### 3.2. Correlation between Pathology and Virtual 3D Image

A scatter plot with linear regression showing the correlation between pathologic tumor size measurement and virtual 3D measurement is presented in Figure 2 and the correlation between pathologic segmentectomy resection margin and virtual segmentectomy resection margin is presented in Figure 3. The tumor size plot showed a positive correlation between tumor size measurements. On the basis of linear regression analysis, the virtual 3D tumor size was estimated as 0.5044 x pathology tumor size (cm) + 0.7506; the Pearson correlation was 0.495 with R2 value of 0.245 (*p* = 0.037). Although the correlation coefficient was not strong, a statistically significant correlation was observed. The plot for resection showed a positive correlation between resection margin distance as measured by the two methods. On the basis of linear regression analysis, the virtual 3D resection margin was estimated as 1.061 x pathology resection margin (cm) + 0.2215; the Pearson correlation was 0.697 with R2 value of 0.4853 (*p* = 0.001). The correlation coefficient was moderate, and a statistically significant correlation was observed.

### 3.3. Factors of Correlation

Univariate and multivariate linear regression analysis of predicative factors contributing to our correlation between pathology report and virtual 3D simulation is shown in Table 2. Parameters analyzed include age, FEV1, Best/Pred (%), FEV1/FVC and resection margin. Both univariate and multivariate analysis on these factors showed 95% confidence intervals crossing zero except in the case of resection margin. Univariate analysis on resection margin yielded OR 0.70, *p* = 0.001, whereas multivariate analysis yielded OR 0.83, *p* = 0.002. Resection margin is a strong predictor. Due to our small sample size, analyses were exploratory in nature.

## 4. Discussion

To our knowledge, this is the first study investigating the correlation between pathology measurement and virtual 3D reconstructed image measurement. Due to the popularity of CT lung carcinoma screening and an increasing trend of pulmonary segmentectomy, thoracic surgeons now face more cases of small tumors with an early disease status. Hence, preserving as maximum lung function with curative intent has become mainstay of treatment. In this retrospective study we correlated pathological measurement with 3D reconstructive figure measurement of small adenocarcinomas in tumor size and segmentectomy resection margin distance. Overall, our results showed that these reconstructed 3D models have a statistically significant positive correlation to tumor size and resection margin.

Tumor size reflects different prognoses and has an impact on therapeutic management [14]. Our study showed that there was a positive correlation between measured tumor size on 3D reconstructed images and pathology, but with its correlation coefficient of 0.245, the correlation was not strong. We noted that 3D image measurements greatly reflect CT measurement since the former is derived from the latter. Hence, our results are similar with previous studies’ finding of poor agreement between radiographic CT measurements and pathological measurements [8,9,15]. Heidinger et al. demonstrate that axial CT diameters were significantly smaller than pathological measurements in the fixed state [8]. However, Lampen–Sachar et al. reported that axial CT measurement was significantly larger than pathological measurement in the fresh state [9]. Moreover, Bhure et al. showed that radiologic axial CT measurements were similar to pathological axial histological section measurements [15]. The variance in our 3D image and pathology size may be attributed to the pathologic specimen section plane possibly not having been accurately aligned with the CT scan images that were used for 3D image reconstruction, and differences in solid and subsolid mass, slice thickness in CT and tissue preparation.

Several techniques have been proposed for the identification of intersegmental planes and anatomical variation during sublobar anatomical resection. Pre-operative planning includes 3D CT bronchography and angiography, virtual-assisted mapping using bronchoscopy multi-spot dye marking and 3D CT [16]. The intersegmental plane can be determined using either inflation–deflation techniques or the popular indocyanine green method [17,18,19]. However, these avenues can only show an intersegmental plane and cannot provide information on the adequacy of surgical margin. An advantage of Ziostation 3D reconstruction imagery is that it can clearly show lung lobes, bronchi, pulmonary arteries and veins. Simulated segmentectomy provides insight for surgeons to evaluate planned approaches and adequate resection margins while being aware of anatomical variations in patients. Figure 1 shows the correlation of CT image and 3D reconstructed image. The 3D model accurately reflects the nodule in its spatial content and the simulated segmentectomy accurately mirrors the surgical specimen in shape, indicating that it is a highly reliable pre-operative planning tool.

Lung cancer has the highest morbidity and mortality rates among malignant tumors worldwide [20]. Nowadays, sublobar resection is considered suitable for early-stage lung cancer and for those with poor cardiopulmonary function. Many studies have confirmed that resection margin status is associated with local recurrence, and a positive margin may present a greater risk for the prognosis of patients with lung cancer [21,22,23]. However, removal of excessive lung tissues may affect postoperative lung function, especially in elderly patients or those with poor lung function prior to surgery [24]. Hence, the delicate balance of sufficient margin distance to ensure a negative margin, without the expanse of post operative pulmonary function lost, is key. Figure 3 shows a strong correlation between virtual 3D model and pathological resection margin. In agreement with our study, pre-operative simulation cases performed by Igai et al. also showcase that Ziosoft software is able to simulate sufficient surgical margin of tumors, which is especially useful in non-palpable tumors [25,26]. Discrepancies between virtual and actual measurements have been reported to be less than 5 mm [26]. This indicates that 3D reconstructed imagery and segmentectomy simulation are simple pre-operative methods that are reproducible, feasible and reliable.

To further investigate factors that may influence the correlation of virtual models and our pathological findings, univariate and multivariate linear regression analyses were performed (Table 2). Our results showed that in both univariate and multivariate analyses, age and pulmonary function outcome did not affect the interrelationship. Resection margin was the sole factor that was statistically significant. This provides further statistical evidence that 3D reconstructed imagery and its operation simulation is a reliable tool in pre-operative planning that is not affected by patients’ age or pulmonary condition. Surgeons can confidently ligate vessels and bronchi in corresponding simulations during segmentectomy to obtain adequate resection margins. This could greatly increase the success of surgical curative intent for early-stage lung adenocarcinoma. Given that no foreseen factors may contribute to its discrepancy, we highly recommend the use of 3D models in pre-operative planning.

During our research process, we found that 3D reconstructed imagery has several advantages with few inconveniences. First, this pre-operative simulation can effectively reduce operation duration and prompt safe operation because of full operative planning from identifying anatomical variations to port placement and simulation of anatomical resection. The images can be juxtaposed during operations to live video-assisted thoracoscopic images to provide guidance and confirmation. Cases that were compared 3D reconstruction model to intraoperative surgical findings were found to have no difference between the anatomy displayed [27,28]. Second, reconstructing vascular imaging on computer is much less invasive with essentially no side-effects or complications compared to conventional diagnostic angiography. Third, the reconstructed image is easy to use and is portable on digital devices. Free space rotation allows visualization from any angle and can help educate doctors about surgical anatomy. Finally, having this virtual model pre-operatively will increase surgeons’ confidence for the planned operation. Drawbacks of 3D CT reconstruction are its cost and the need for manual reconstruction prior to surgery.

We acknowledge that this study had limitations and bias. First, our study was retrospective, so selection bias was inherent to our study design given that only surgically resected and pathology proven adenocarcinomas were included. Second, the section thickness of the evaluated CT images was not always the same so the precision of reconstructed images may be affected. Third, variations in size measurements between observers in the pathologic evaluation may have an effect our results. As the nearest resection margin is difficult to evaluate upon tumor excision and pathological processing, this is also a limitation. Retrospective selection of cases also resulted in a small sample size, and more studies are required to confirm clinical applicability.

## 5. Conclusions

The 3D display of anatomic structures can be extrapolated to predict pathology tumor size and resection margin in simulated segmentectomy of early-stage pulmonary adenocarcinoma. This tool aids our understanding of patient anatomy, prompting confidence in pre-operative planning and ensuring successful segmentectomy outcomes. Regardless of age and pulmonary function, 3D reconstructive imagery and operation simulation can provide accurate outcomes, marking it as a simple, effective and feasible pre-operative planning tool.

## Figures and Tables

**Figure 1 jcm-11-06155-f001:**
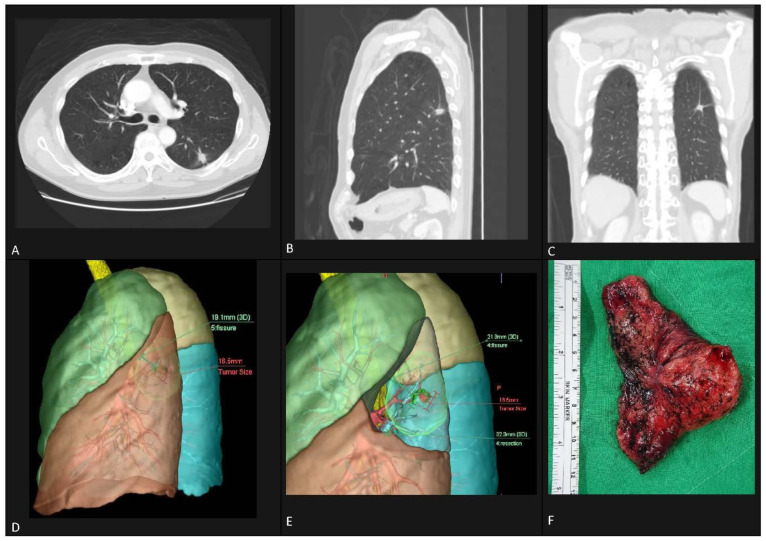
Pre-operative computed tomography lung window in transverse view (**A**), coronal view (**B**) and sagittal view (**C**) revealed an irregular contour 18-mm part solid nodular shadow in the left lower lobe posterior region. Simulated imaging from pre-operative chest CT created by Ziostation2 software revealed the nodule location in relation to surrounding vessels and fissures. The tumor is in segment 6 with an elevated segment 10 anatomy (**D**). Simulated left S6 segmentectomy, showing surgical margin between the tumor and its closest boundary (**E**). Gross appearance of left S6 segmentectomy (**F**).

**Figure 2 jcm-11-06155-f002:**
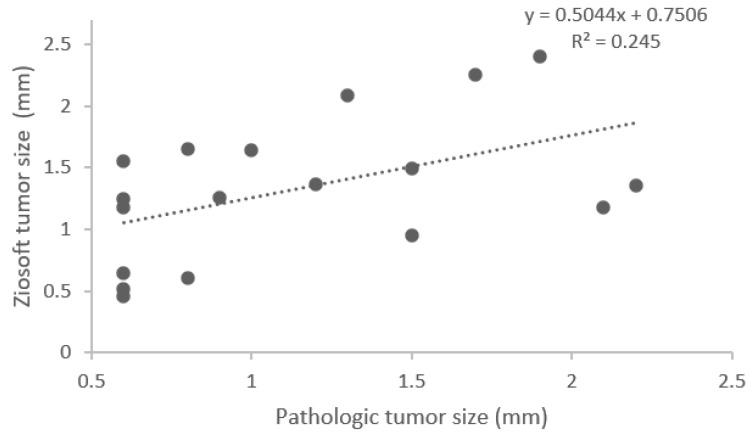
Linear regression analysis showing correlation between pathologic tumor size measurement and virtual 3D tumor size measurement.

**Figure 3 jcm-11-06155-f003:**
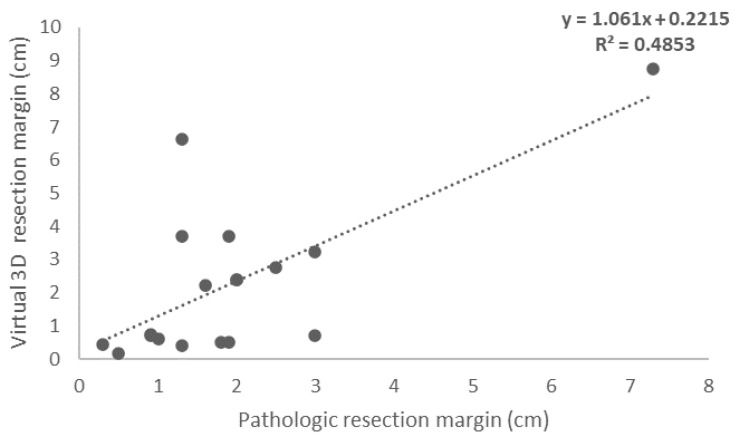
Linear regression analysis showing correlation between pathologic segmentectomy resection margin and virtual segmentectomy resection margin.

**Table 1 jcm-11-06155-t001:** Demography of patients undergoing segmentectomy.

Parameter	N	%
**Total patients**	18	100%
**Demographics**		
Age, median (IQR)	60 (47.50–65.25)	
Male	7	38.9
Female	11	61.1
**Underlying Disease**		
Diabetes	4	22.2
Hypertension	5	27.8
Smoker	2	11.1
COPD	1	5.6
**Pulmonary Function (median)**		
FEV1	2.40 (1.91–2.62)	
Best/Pred (%)	89.20 (81.75–105.15)	
FEV1/FVC	78.88 (74.80–85.23)	
**Tumor location**		
Left	7	38.9
Right	11	61.1
**Histopathology**		
Acinar	9	50
BAC/lepidic	4	22.2
Papillary	3	16.6
Minimal invasive	2	11.1
**Tumor Size, median (IQR)**		
Pathology measurement	1.31 (0.87–1.64)	
Virtual 3D measurement	0.95 (0.60–1.55)	
**Resection Margin, mean (IQR)**		
Pathology measurement	1.48 (0.50–3.35)	
Virtual 3D measurement	1.7 (0.98–2.12)	
**Length of hospital stay (day)**	5 (5–7)	

3D, three-dimensional; BAC, bronchioloalveolar carcinoma; COPD, chronic obstructive pulmonary disease; FEV1, forced expiratory volume in one second; FVC, forced vital capacity; IQR, interquartile range; Pred, predicted.

**Table 2 jcm-11-06155-t002:** Univariate and multivariate linear regression analysis of factors that may contribute to the correlation of pathology report and virtual 3D simulation.

Parameters	Univariate			Multivariate		
	OR	95% CI	*p* Value	OR	95% CI	*p* Value
Age	−0.01	−0.07–0.06	0.979	0.13	−0.05–0.08	0.615
FEV1	0.32	−0.45–2.00	0.200	0.29	−0.83–2.26	0.333
Best/pred (%)	−0.01	−0.03–0.03	0.966	−0.25	−0.07–0.04	0.588
FEV1/FVC	−0.04	−0.04–0.04	0.880	0.63	−0.03–0.13	0.194
Resection margin	0.70	0.21–0.71	0.001 *	0.82	0.23–0.84	0.002 *

* Significant as *p*-value <0.05.

## Data Availability

All data are available upon reasonable request.

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
