# Peer review of "Size Measurement and Segmentectomy Resection Margin of Early-Stage Lung Adenocarcinoma Manifesting on Virtual 3D Imagery and Pathology: A Pilot Correlation Study"

_jcm, 2022, doi:10.3390/jcm11206155_

Round 1

Reviewer 1 Report

The work shows the application and combination of computer science and surgery. The work is clear and well prepared. Figure A-C (CT image) could suggest segment VI tumor. The 3-D Ziostation2 reconstruction revealed a small segment VI and elevated segment X anatomy, which is crucial for the surgeon.

Author Response

We thank the reviewer for the time and effort invested into the review of our manuscript and for the helpful comments and suggestions. We have made the change to Figure 1 legend to cooperate this specific finding.

Reviewer 2 Report

Thank you for the opportunity of reviewing this article.

It was well written and a pleasure to review.
I agree the idea of this study is clinically relevant. There are a few comments that require further consideration to better understand the translation of these results.

In the 3D simulation measurement section(Materials and Methods 2.2), authors described that the tumor boundary was manually edited to excluded peri-nonsolid region. Segmentectomy is often indicated for adenocarcinoma showing ground glass opacity (GGO) as most of them are non-invasive. Authors should mention why they excluded GGO areas of tumor when they set the surgical margins.

In the Pathology tumor size and resection margin section (Materials and Methods 2.4), resection margin was measured as the nearest margin. I wonder how authors ensure they found the ‘true’ nearest margin as it would be so difficult to check all directions of the specimen once the resected tumor was cut. Did the pathologist calculate the all direction-margins for all slides during the cutting  process or did the authors examine all the specimens again to check the recorded margin was the true tumor margin? If not, authors should mention this as a limitation.

In Table 1, I suggest to describe the variables using the median and IQR rather than mean and SD unless the variables were normally distributed.

Table 2 appears to have many problems to me.

The explanation “Correlation of pathology report and virtual 3D “stimulation” ( a typo of “simulation”?) was unclear to me. Basically, is it statistically valid to perform the uni-and multi-variate analysis (five covariables) when the patient cohort is as small as 18? The results were self-evident to me because the 3D method did not depend on age, sex, or lung shape.

In the first paragraph of the discussion part, authors described that 3D model is applicable to patients to all ages and pulmonary function conditions. I do not think you can conclude this because the analysis invalid as mentioned above. Authors should add data if they are to mention that their 3D techniques are applicable, regardless of all age and pulmonary function.

In the third paragraph of the discussion part, authors described that the simulated segmentectomy ca accurately mirror the surgical specimen in shape. However, I wonder how authors ensured that the authors resected the lung as the Ziostation 2 planed (tumor size/location-based 3D simulation), while authors resected the lung using a inflation/deflation method. I suggest authors to elaborate this very carefully because the concept of the current retrospective study should have been evaluated prospectively. I didn’t think the current description was persuasive enough. 

In the reference section, are ref #25 and 26 the same ?

Round 2

Reviewer 2 Report

Authors have responded to my all questions. I don't have further questions.